# The Social Values of Nursing Staff and the Perceived Quality of Their Professional Lives

**DOI:** 10.3390/healthcare11202720

**Published:** 2023-10-12

**Authors:** Francisco Javier Mazuecos, Ángel De-Juanas Oliva, Ana Eva Rodríguez-Bravo, Javier Páez Gallego

**Affiliations:** 1Faculty of Education, Universidad Nacional de Educación a Distancia, Calle Juan del Rosal 14, 28040 Madrid, Spain; fjmazuecos.ts@gmail.com (F.J.M.); anaeva.rodriguez@edu.uned.es (A.E.R.-B.); 2Faculty of Law, Universidad Nacional de Educación a Distancia, Calle del Obispo Trejo, 2, 28040 Madrid, Spain; javier.paez@der.uned.es

**Keywords:** social values, nursing staff, professional quality of life, life satisfaction, social behaviour

## Abstract

This study’s main purpose involves exploring the relationship between the social values of nursing staff and the perception they have of their professional lives. A further aim is to examine how their terms of employment and tenure of service relate to the quality of their careers and their social values. The research consisted of a non-experimental quantitative approach of a descriptive nature involving 380 nursing staff at four public hospitals in Madrid (Spain). The values were appraised by means of the Schwarz Value Survey (SVS) and the quality of their careers was measured through the Quality of Professional Life (QPL-35) questionnaire. The results reveal significant correlations between the two, highlighting the significance of such values as universalism, benevolence, achievement and power depending on their terms of employment, on the one hand, and all the values in the Schwartz model according to the length of their tenure on the other. The findings suggest that terms of employment and tenure are significantly related to the axiological profile of nursing staff and the quality of their professional lives. This study provides major empirical evidence that contributes to our understanding of how social values and the quality of professional lives are interwoven within the field of nursing in Spain.

## 1. Introduction

Professional quality of life (QPL), understood as the satisfaction of a set of needs of a worker through resources, activities and outcomes derived from participation in the workplace, is a fundamental concern in healthcare, and specifically for nurses [1,2,3], not only because of its impact on the health and wellbeing of the nurses themselves [4], but also through its influence on the overall quality of care within the ambit of health services [5,6,7]. Within the broad and dynamic field of nursing, QPL has become an issue of growing research interest, especially in the wake of the COVID-19 pandemic [8,9,10,11,12,13]. The nursing profession is crucial for people’s health and wellbeing and now faces a series of challenges and demands that are constantly evolving as medical care develops in response to advances in technology, as well as to ongoing demographic and social changes [14,15,16,17]. These changes may have a direct impact on the QPL of nursing staff that perform a key role by caring for patients of all ages and conditions. A vital step towards a better understanding of the factors that underpin nurses’ QPL involves considering the influence their social values have on their day-to-day care of patients [18,19]. This context has provided the setting for a number of studies that consider the impact that value systems and ethics have on nursing staff [5,20,21,22,23,24,25], but to the best of our knowledge there has still not been any investigation that relates their system of social values and their QPL. A good quality of professional life is considered fundamental because of its inescapable contribution to professionals feeling competent, autonomous and linked to the vision and mission of the organisation and of their profession, and consequently, that they have covered the basic psychological needs that the Self-Determination Theory marks as universal because they are present in every culture [26,27,28,29].

This study is therefore based on the Schwartz model because it provides an appropriate theoretical model for examining how individual values may interact with the professional duties of nursing staff. This link between the values of Schwartz’s universal model and nursing is evident in various aspects of the profession. On the one hand, in the more informal dimension, this is clear in the cultural awareness that has been generated of this profession and of the values associated with this culture; these values are assimilated by the professionals who practise it. On the other hand, in the formal dimension, these values are included in different formulations but with similar intentions in the codes of ethics of professional associations and professional bodies at the national and international level. Therefore, the pro-social nature of the profession is linked to this pole of values in Schwartz’s model [30].

Furthermore, the Schwartz Value Survey [31] provides a widely accepted instrument for understanding the basic and universal dimensions of human values. Values may be defined as real qualities or lasting and relatively stable beliefs that inform people’s choices and actions in different situations and moments in their lives [32,33]. Along these lines, Schwartz [34,35] identifies a structure of ten personal values, each one with its own set of distinctive characteristics. These values are conformity, tradition, security, power, achievement, hedonism, stimulation, self-direction, universalism and benevolence. The interaction and prioritisation of these values constitute the way people perceive the world, make decisions and relate to each other. In the case of nursing, a field which involves care and support, it is intrinsically linked to the core values of benevolence and universalism identified by Schwartz, highlighting the concern for the wellbeing of others, safeguarding human dignity and championing equality [20]. Nursing staff are often immersed in situations that call for empathy, compassion and a genuine commitment to their patients’ wellbeing [36,37]. In addition to the value of self-direction implied by nurses’ independence and freedom in their duties, they provide the framework for teamwork with other health professionals in the pursuit of a purpose that goes beyond personal satisfaction and is deeply rooted in the vocation that underpins the nursing profession [38]. Nevertheless, the very nature of this profession may bring it into conflict with some of the personal values held by nursing staff, and this may impact their professional lives. For example, nursing may cause stress due to a high workload, a lack of resources or difficult decision-making in challenging situations in which patients may be in pain or suffering. Nurses may also be affected by high demands in terms of time and energy that, in turn, may lead to tensions between their professional dedication and their own personal satisfaction and wellbeing [39,40]. In short, nursing staff may have to cope with the difficult challenge of striking a balance between their professional duties and their personal needs and expectations, which may in turn affect their perception of their QPL.

The terms of employment of nursing staff may also have a bearing on their QPL and their emotional wellbeing [41]. Secure terms of employment may help to improve their job satisfaction and their feelings of engagement; by contrast, insecure or precarious employment may lead to discontent and negatively impact their QPL and job satisfaction [42,43]. Elsewhere, the organisation of work, ergonomics and safe and healthy working conditions that consider their needs and demands; the existence of autonomous competencies and perceived independence; and a management team that stimulates the development and improvement of their duties have also been singled out as factors associated with the QPL of nursing staff [4,9,44].

Against this background, the main aim here is to analyse the interaction between the social values of nursing staff and their perceived QPL, as well as to study the interaction between their temporary, replacement and full-time employment and their tenure in terms of their QPL and social values.

## 2. Materials and Methods

In keeping with the established aims, this study was conducted based on a quantitative method of a descriptive, non-experimental and ex post facto nature that considered the possible relationships between the QPL and axiological profile of Spanish nursing staff. The following hypotheses were formulated for this study:

**H1.** *There is a relationship between the QPL of nursing staff and their social values*.

**H2.** *The terms of employment of nursing staff are related to their QPL and their axiological profile*.

**H3.** *Tenure of service among nursing staff is related to their QPL and their social values*.

### 2.1. Participants

The sample consisted of 380 active nurses in Madrid (Spain) (males, n = 70, 16.8%; females, n = 310, 73.7%). The cohort was chosen using opportunity sampling according to criteria of access and availability. The participants were recruited from four public hospitals located in different parts of the city. Three groups were formed according to their terms of employment (The current employment contracts in this case in Spain are as follows: *fijos* are indefinite, open-ended, and full-time; *Interinos* are hired to fill a gap in the workforce for the time the person they replace is away because of, for example, maternity and sick leave; and, finally, *eventualees* are temporary employees hired for a specific length of time): (1) full-time contracts (n = 222; 58.42%); (2) replacement contracts (n = 22; 21.84%); and (3) temporary contracts (n = 75; 19.74%). Five groups were formed for tenure: (1) fewer than five years’ experience (n = 30; 8.82%); (2) between 6 and 12 years (n = 55; 16.16%); (3) between 13 and 19 years (n = 70; 20.59%); (4) between 20 and 26 years (n = 77; 22.65%); and (5) more than 27 years (n = 108; 31.76%). Any nurses that had been working for under a year were excluded from the analyses related to H3.

### 2.2. Materials

The data on sociodemographic details, terms of employment and tenure were gathered through an initial survey prior to the application of the two measuring instruments used to collect information and respond to the study’s main purpose. The first instrument was the Schwartz Value Survey (SVS) [45], adapted for the Spanish population by Páez and De-Juanas [46]. This survey measures participants’ preferred values according to the Schwartz Theory of Basic Human Values. The SVS consists of 45 items whose scores are grouped into the ten universal values that people find motivational, which are as follows: *self-direction*: this value is related to independent thinking and decision-making, with a leaning towards creativity, freedom, choosing one’s own goals, curiosity and independence; *stimulation*: a value whose main characteristic is self-affirmation, which is related to the search for stimuli and life challenges, a varied life style and a spirit of adventure; *hedonism*: this entails a personal quest for satisfaction through positive emotional experiences; *security*: This is related to a desire for personal stability within both the personal and social milieus. The outcome of embracing this value involves prioritising the achievement of macro-, meso- and microsocial stability, as well as forging positive interpersonal relationships through a generally accepted social order. This value indicates an aspiration for individual and group survival; *conformity*: This value involves refraining from inconveniencing or upsetting others by upholding those social values that protect their freedom and respect. This value’s main features are obedience, self-discipline and courtesy; *tradition*: This value refers to respect for the customs and ideas determined by traditional culture and norms. This value entails attitudes of respect for tradition, religion, subordination and moderation; *achievement*: This value involves the search for social recognition and approval according to social norms. This value encompasses attitudes such as ambition and influence over others; *power*: this refers to the interest in prioritising control and dominance over others, either individually or over groups that belong to the same ambit in which the person lives; *benevolence*: this value is linked to the need to uphold positive relationships for maintaining the wellbeing of both individuals and groups, and is linked to attitudes such as support, honesty, forgiveness, loyalty and responsibility; and, finally: *universalism*: this value is defined by the survival instinct that individual and groups manifest when the resources they depend on are threatened. This value encompasses such attitudes as personal maturity, tolerance, wisdom, environmental protection, social justice, general social wellbeing and equality. This instrument recorded a Cronbach’s alpha of internal consistency of 0.90 (α = 0.90).

The second instrument was the Quality of Professional Life (QPL-35) questionnaire associated with Karasek’s demand–control model [45,47], validated by Cabezas [48] and Martín et al. [49] for the Spanish population. This scale measures the overall quality of professional life and includes employment aspects such as health, stress levels, the necessary training and the burden of responsibilities. The scale provides an overall score that ranges between 1 and 350 points. In turn, this instrument recorded a Cronbach’s alpha of 0.89 (α = 0.89).

### 2.3. Procedure

The instruments were administered via a link to an online form uploaded in the Survey Monkey application (platform for online surveys). The participants were informed of the study’s purpose, and they took part on a voluntary basis according to the instructions received. The participants did not receive any financial compensation.

### 2.4. Data Analysis

The first step involved a descriptive statistical analysis, followed by analyses of correlations through the Pearson correlation coefficient (r). The final step involved an analysis of variance. All the steps were carried out with IBM^®^ SPSS version 29.0 software for Macintosh.

## 3. Results

### 3.1. Descriptive Analyses of the Nursing Staff’s System of Values and Their QPL

As regards the sample’s preferences, a study was conducted for each one of the ten value domains in the model. Table 1 presents the results.

The participants’ highest value preferences involved the domains of *benevolence*, *self-direction* and *universalism*. They all recorded a mean score of more than seven points (Table 1).

By contrast, the values recording the lowest means in terms of preference were *tradition* and *power*, both scoring under 6.1 points.

All the values recorded a dispersion ranging between σ = 2.0 and σ = 1.76, which may be considered small as the scores were clustered around the mean scores of the axioms.

The sample’s overall QPL variable returned a mean of 234.73 points ($\bar{x}$ = 234.73), which places this figure above the mean point on the scale’s range. Nevertheless, this score’s standard deviation reveals that the responses are widely scattered (σ = 35.99).

### 3.2. Correlation Analysis

#### 3.2.1. Relationship between the QPL of Nursing Staff and Their System of Values

With a view to confirm H1 on the relationship between the QPL and axiological profile of nursing staff, a correlation analysis was conducted between the scores for the axioms and the overall score for QPL. A positive correlation means that a high score in the values is accompanied by a high score in QPL, and vice versa. In turn, a negative correlation means that a high score in either one of the two crossed variables is reflected in a low score in the other one. Table 2 shows the results.

The ten axioms in the Schwartz model recorded a high and positive correlation with a high QPL among nursing staff, with all the correlations recording a probability of less than 0.01 (*universalism*, r = 0.224, *p* = 0.000; *benevolence*, r = 0.234, *p* = 0.000; *conformity*, r = 0.199, *p* = 0.000; *stimulation*, r = 0.256, *p* = 0.000; *hedonism*, r = 0.221, *p* = 0.000; *achievement*, r = 0.268, *p* = 0.000; *power*, r = 0.366, *p* = 0.000; *tradition*, r = 0.285, *p* = 0.000; *security*, r = 0.260, *p* = 0.000; and *self-direction*, r = 0.238, *p* = 0.000).

#### 3.2.2. Relationship between the QPL and Axiological Profile of Nursing Staff Depending on Their Terms of Employment

The nursing staff with temporary contracts recorded significant correlations between QPL and the values of *tradition* (r = 0.250, *p* = 0.040), *achievement* (r = 0.257, *p* = 0.034) and *power* (r = 0.396, *p* = 0.001) (Table 3). These last two values define an axiological profile called *self-promotion*.

In turn, the nurses with replacement contracts recorded three significant correlations. Two of these were negative, whereby high values in these values are related to significantly low ones in QPL, and vice versa: *universalism* (r = −0.281, *p* = 0.015) and *benevolence* (r = −0.357, *p* = 0.002). The third was positive for the value *power* (r = 0.419, *p* = 0.000).

Finally, the nurses with full-time jobs recorded two significant correlations between QPL and the values of *achievement* (r = 0.150, *p* = 0.035) and *power* (r = 0.173, *p* = 0.015).

The groups share the significance of the relationship of *power*, with a high correlation and probability of less than 0.01 in all three cases.

#### 3.2.3. Relationship between QPL and Axiological Profile According to Tenure

The two variables were crossed in order to verify the hypotheses on the relationship between the variables QPL and the value domains according to the nurses’ terms of employment (Table 4). This analysis discarded the 36 participants with a tenure of less than a year. Overall QPL was measured against the ten values in the Schwartz model according to the participants’ tenure to discover whether there is a relationship between the groups formed by this demographic variable.

The group with fewer than five years’ tenure recorded two significant correlations between the values *security* (r = 0.381, *p* = 0.042) and *self-direction* (r = 0.422, *p* = 0.023) and QPL, both of which were positive. These values are distant from each other in the Schwartz model and do not correspond to any specific axiological model. This demographic group recorded the most significant correlations.

In turn, the group of nursing staff with a tenure of 6–12 years recorded three significant correlations for the values of *tradition* (r = 0.371, *p* = 0.005), *power* (r = 0.417, *p* = 0.002) and *hedonism* (r = 0.281, *p* = 0.018). Nevertheless, these three values did not cluster to form recognisable axiological profiles.

The group whose tenure ranged between 13 and 19 years recorded by far the highest number of significant correlations, ahead of the other four. It recorded significant correlations with the values *universalism* (r = −0.272, *p* = 0.023), *benevolence* (r = −0.335, *p* = 0.005), *conformity* (r = −0.381, *p* = 0.001), *hedonism* (r = −0.367, *p* = 0.002), *stimulation* (r = −0.331, *p* = 0.005) and *self-direction* (r = 0.338, *p* = 0.004). It is important to note that all these correlations were negative, which means that a high score in overall QPL reflects a weak identification with these axioms, and vice versa.

These six axioms are grouped in the higher-order values of *self-transcendence* and *openness to change*, thereby defining two specific axiological profiles that explain the differences in overall QPL.

Likewise, those nurses whose tenure ranged between 20 and 26 years recorded two significant correlations with the values of *achievement* (r = 0.337, *p* = 0.002) and *power* (r = 0.288, *p* = 0.011). Both axioms constitute the higher-order value of *self-promotion.*

Finally, the group with a tenure equal to or more than 27 years recorded three statistically significant correlations between overall QPL and the axioms of *achievement* (r = 0.263, *p* = 0.005), *power* (r = 0.279, *p* = 0.003) and *stimulation* (r = 0.239, *p* = 0.011). As in the 20–26 group, the first two values are grouped in the higher-order value *self-promotion.*

An analysis of these relationships according to the groups formed when the sample is classified by “Terms of Employment” reveals that nursing staff with temporary contracts have a greater preference for the value *conformity*, like those with replacement contracts. Those nurses on full-time contracts, however, feel more secure and are more motivated by *achievement* and *self-direction*, with the latter involving independence of action and thought, for example.

Elsewhere, the relationship between the values and the sundry groups formed by the classifying variable “Tenure” reveals differences in value preferences in the sample. The nursing staff with fewer than five years’ tenure focus more on *achievement* and *security* than on *self-direction*, whereas those with 6–12 years’ tenure prefer *conformity*.

### 3.3. Analysis of Inferences

The aim here was to discover any significant differences in the nurses’ overall QPL depending on their terms of employment and tenure.

#### 3.3.1. Differences in QPL According to Terms of Employment

An analysis of the differences in overall QPL according to terms of employment records statistically significant results. The descriptive values for the corresponding QPL are presented in Table 5.

The probability associated with the F value was less than 0.05 (F3.376 = 2.522; *p* = 0.048) (Table 6).

A subsequent verification of the analysis of variances revealed that differences arose between nursing staff with a full-time contract and those with a temporary one (Tukey HSD post hoc test = −23.85; *p* = 0.005)

#### 3.3.2. Differences in QPL Depending on Tenure

Tenure prompted significant differences in the scores nursing staff gave to overall QPL. Table 7 shows each group’s descriptives according to tenure. This analysis discarded 36 participants with less than one year’s tenure.

The probability associated with the F value was less than 0.05 (F5.374 = 2.520; *p* = 0.029), as Table 8 shows.

A significant difference was recorded between the group with fewer than five years’ tenure and the group with 27 years’ or more, as shown by Tukey’s post hoc test (Table 9).

## 4. Discussion

As medical care continues to advance and face ever-greater challenges, the relationship between nurses’ QPL and their social values is a complex field of research that calls for more attention. This study’s findings reveal that the system of motivational values among nursing staff focuses more on a set of values related to the development of human relationships, fairness and respect [50,51,52,53]. The vocation for care and support that underpins nursing often involves deeply rooted social values, such as responsibility, integrity and compassion. These core values may have an influence on the choice of the nursing profession and on the dedication that nursing staff apply to their day-to-day duties. Social values related to equality, fairness and human dignity may prompt nurses to provide all their patients with equitable and consummate care; in short, these are values that help to safeguard patients’ dignity and provide a quality service. Our study’s results are consistent with those of other studies [20,21,22,24,38,54,55].

Furthermore, the terms of employment of nursing staff may also impact their job satisfaction and their emotional wellbeing. Secure working conditions may improve their job satisfaction and their sense of engagement. In turn, temporary and precarious employment may lead to discontent and have a negative effect on their QPL. Their QPL may likewise be eroded by factors such as the absence of measures that favour a better workplace environment in terms of safe and healthy ergonomic conditions, policies, programmes and actions designed to respond to their needs and expectations regarding their workload, and a lack of autonomous competencies, perceived independence and managerial support for undertaking and improving their duties [4,44,56].

This study’s findings also reveal that nurses’ systems of values and their QPL are influenced by their tenure. A study by Şenyuva [23] involving a sample of 700 nursing staff of different ages in Istanbul reports that all the nurses across the generations prioritise human dignity and responsibility among their profession’s social values. Nevertheless, compared with their colleagues in older age groups, young nurses give preference to these values over others such as equality, fairness and freedom. Likewise, Basit et al. [19] find a relationship between nurses’ tenure and their perception of values. In our study, nursing staff with a shorter tenure correlate the values of *power* and *tradition* with QPL, as opposed to other groups with more experience in which *achievement* and *power* together with *stimulation* correlate to a greater extent in a statistically significant manner. Finally, it should be noted for those nurses with a longer tenure, but still less than twenty years, that the values of *universalism* and *benevolence* correlate negatively with QPL. This may explain why social values may also create challenges and tensions for nurses’ QPL. Time demands and the need to uphold high standards of care may lead to stress and burnout, which may in turn affect their job satisfaction, their emotional wellbeing and their treatment of patients [57,58,59,60].

## 5. Conclusions

This study’s results reveal that the QPL of nursing staff is a core component in the provision of quality medical care [19] and in the promotion of their wellbeing. Social values play a key role in the way nurses experience their jobs and how they relate to their working environment. Our findings have highlighted a more profound understanding of nurses’ experience of their workplace and, ultimately, of the medical care they provide for their patients by confirming H1 and showing that the participants’ overall QPL correlates with all the domains in the systems of values Schwartz propounds.

As regards H2, terms of employment interacts positively with nurses’ QPL and their values, with the exception of *achievement* and *power*, which do not record a positive relationship with QPL.

H3 is also confirmed, revealing that nurses’ tenure is related to their QPL and their system of values. In their first years in nursing, they embrace the values of *power* and *tradition*, subsequently focusing on *self-direction* and, as the years go by, to a greater extent on *achievement, power* and *stimulation*. Understanding the relationship between nurses’ social values and tenure may be useful for designing effective strategies that lead to the best possible QPL for this professional cohort.

The application of the Schwartz Values System as our starting point enables us to organise the complex interactions between nurses’ social values and their QPL. This study has set out to further a more profound understanding of these interrelated aspects and provide valuable knowledge for improving the nursing profession and the health and wellbeing of nursing staff.

Finally, regarding the study’s limitations and future research, the data presented in this article may provide a reference framework for the sample in question. Nevertheless, although the sample is very large, the results provide a specific snapshot that limits its generalisation, whereby it would be expedient to replicate this study in other populations. Furthermore, we are aware that the Schwartz model of motivational values provides a general view of the universal values that inform people’s behaviour. This means that this model can be used to understand the way nursing staff function, but only in a general sense. This model could therefore be complemented by other measures that focus on nurses’ working values [38], with this contextual approach adding a further perspective. We consider this to be of interest for future studies. We also agree with Whyle and Olivier [5] regarding the need to apply social science methods to conduct a more thorough empirical study of nurses’ social values. We therefore consider that our research could be enriched by another study of a qualitative nature that addresses the interaction between social values and QPL from a mixed-method approach, which would allow one to interpret the results by counting on the nurses’ own testimonies.

## Figures and Tables

**Table 1 healthcare-11-02720-t001:** Value preferences.

	N	M	SD
Universalism	380	6.9853	1.76
Benevolence	380	7.4211	1.96
Security	380	6.8296	1.76
Conformity	380	6.9303	1.80
Tradition	380	6.0908	1.88
Achievement	380	6.4711	2.00
Power	380	5.4474	1.90
Hedonism	380	6.6947	1.95
Stimulation	380	6.4566	1.89
Self-direction	380	7.0123	1.82

M = mean; SD = standard deviation.

**Table 2 healthcare-11-02720-t002:** Correlation between QPL and the value domains.

		Self-Direction	Benevolence	Conformity	Stimulation	Hedonism	Achievement	Power	Security	Tradition	Universalism
QPL	Pearson’s r	0.238 *	0.234 *	0.199 *	0.256 *	0.221 *	0.268 *	0.366 *	0.260 *	0.285 *	0.224 *
	Sig (2-tailed)	0.000	0.000	0.000	0.000	0.000	0.000	0.000	0.000	0.000	0.000
	N	380	380	380	380	380	380	380	380	380	380

* *p* < 0.05.

**Table 3 healthcare-11-02720-t003:** Correlation between QPL and the value domains according to the nurses’ terms of employment.

QPL		Universalism	Benevolence	Security	Conformity	Tradition	Achievement	Power	Hedonism	Stimulation	Self-Direction
Temp	Pearson’s r	−0.032	0.067	0.162	−0.068	0.250 *	0.257 *	0.396 **	0.145	0.117	0.226
	Sig (2-tailed)	0.796	0.587	0.187	0.583	0.040	0.034	0.001	0.238	0.343	0.064
	N	68	68	68	68	68	68	68	68	68	68
Replacement	Pearson’s r	−0.281 *	−0.357 **	−0.071	−0.219	0.176	0.182	0.419 **	−0.098	−0.040	−0.226
	Sig (2-tailed)	0.015	0.002	0.546	0.061	0.133	0.122	0.000	0.405	0.732	0.052
	N	74	74	74	74	74	74	74	74	74	74
Full-time	Pearson’s r	0.002	−0.022	−0.040	−0.039	0.045	0.150 *	0.173 *	0.008	0.079	−0.037
	Sig (2-tailed)	0.981	0.758	0.576	0.585	0.523	0.035	0.015	0.907	0.269	0.603
	N	198	198	198	198	198	198	198	198	198	198

* *p* < 0.05. ** *p* < 0.01.

**Table 4 healthcare-11-02720-t004:** Correlation between QPL and the value domains according to the nurses’ tenure.

QPL		Universalism	Benevolence	Security	Conformity	Tradition	Achievement	Power	Hedonism	Stimulation	Self-Direction
<5 years	Pearson’s r	0.213	0.267	0.381 *	0.149	0.141	0.274	0.113	0.117	0.312	0.422 *
	Sig (2-tailed)	0.268	0.161	0.042	0.441	0.466	0.150	0.561	0.547	0.099	0.023
	N	30	30	30	30	30	30	30	30	30	30
6–12 y	Pearson’s r	−0.135	−0.044	0.180	0.033	0.371 **	0.200	0.417 **	0.281 *	0.100	0.164
	Sig (2-tailed)	0.327	0.751	0.188	0.813	0.005	0.143	0.002	0.038	0.467	0.233
	N	55	55	55	55	55	55	55	55	55	55
13–19 y	Pearson’s r	−0.272 *	−0.335 **	−0.227	−0.381 **	0.081	−0.100	0.117	−0.367 **	−0.331 **	−0.338 **
	Sig (2-tailed)	0.023	0.005	0.059	0.001	0.505	0.412	0.335	0.002	0.005	0.004
	N	70	70	70	70	70	70	70	70	70	70
20–26 y	Pearson’s r	0.038	0.077	0.064	0.098	0.104	0.337 **	0.284 *	0.021	0.202	0.042
	Sig (2-tailed)	0.739	0.498	0.573	0.389	0.362	0.002	0.011	0.853	0.075	0.715
	N	77	77	77	77	77	77	77	77	77	77
>27 y	Pearson’s r	−0.002	−0.058	−0.023	−0.035	−0.044	0.263 **	0.279 **	0.156	0.239 *	−0.009
	Sig (2-tailed)	0.983	0.544	0.810	0.717	0.644	0.005	0.003	0.103	0.011	0.923
	N	108	108	108	108	108	109	108	109	108	108

* *p* < 0.05. ** *p* < 0.01.

**Table 5 healthcare-11-02720-t005:** Descriptive statistics for QPL for each type of employment status.

Value Preferences			
	N	M	SD
Replacement staff	83	235.4595	33.06456
Full-time staff	222	236.8894	32.31252
Temporary staff	75	238.8971	28.41964
Total	380	234.7289	35.99337

**Table 6 healthcare-11-02720-t006:** Analysis of variance of QPL according to terms of employment.

	Sum of Squares	df	Root Mean Square	F	Sig.
Inter-groups	31.931	3	10.644	2.522	0.048
Intra-groups	1587.006	376	4.221		
Total	1618.937	379			

**Table 7 healthcare-11-02720-t007:** Descriptive statistics for QPL depending on tenure.

	N	M	SD
Fewer than 5 years	29	243.0000	28.86174
6–12 years	55	238.4000	32.34398
13–19 years	70	228.3857	31.91033
20–26 years	79	235.4304	31.13811
27 years or more	111	241.1081	31.34278
Total	344	234.7289	35.99337

**Table 8 healthcare-11-02720-t008:** Comparison of measures of QPL according to tenure.

	Sum of Squares	df	Root Mean Square	F	Sig.
Inter-groups	52.756	5	10.551	2.520	0.029
Intra-groups	1566.181	374	4.188		
Total	1618.937	379			

**Table 9 healthcare-11-02720-t009:** Post hoc test of the difference in means between groups according to tenure.

Tenure		Difference in Means	Sig.
27 years or more	Fewer than 5 years	1.15841 *	0.039
	6–12 years	−0.40385	0.934
	13–19 years	0.50336	0.670
	20–26 years	0.30206	0.928
	20–26 years	0.23823	0.969

* *p* < 0.05.

## Data Availability

The data presented in this study are available on request from the corresponding author.

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
