# Peer review of "The Social Values of Nursing Staff and the Perceived Quality of Their Professional Lives"

_healthcare, 2023, doi:10.3390/healthcare11202720_

Round 1

Reviewer 1 Report

In relation to the interest of study.
Novelty and opportunity.
Significant contributions of the research related to the attempt to determine the predominant values in health care and their potential change (specifically in the field of nursing) based on two major variables (1) employment status that speaks of stability or instability in employment and (2) seniority in the position. These variables are determinant and well related to the type of study; do the values change before specific situations linked to stability and seniority in the position? What relationship do they have with the quality of working life?
The research questions are logical and are well specified in objectives and working hypotheses.
The quantitative methodology used, although not new, is adequate.
The results of the study demonstrate the need for further research on the dimensions of values and quality of life as well as their impact on performance and quality of care in the field of health care.

State of the question

In the theoretical field, an important limitation is detected: it is necessary to explain what is understood by quality of professional life (explicit theoretical model and dimensions of analysis), as well as to better justify the relationship between quality of professional life and possession of values. It is advisable to better justify the relationship of the values assumed as proper to nursing with the proposed model of the universal values of Schwarz (they allude to culture, cognitive sphere of the person that impacts on his/her behavior).
References that we can suggest

Wauters, M., Zamboni Berra, T., de Almeida Crispim, J., Arcêncio, R. A., & Cartagena-Ramos, D. (2023). Calidad de vida del personal de salud durante la pandemia de COVID-19: revisión exploratoria. Revista Panamericana de Salud Pública, 46, e30.

Vizcaíno, Y. Y. M., Alonso, M. D. C. V., & Vizcaíno, Y. M. (2020). Factores involucrados en la calidad de vida laboral para el ejercicio de la enfermería. Revista Cubana de Medicina Militar, 49(2), 364-374.

Castro, P., Cruz, E., Hernández, J., Vargas, R., Luis, K., Gatica, L., & Tepal, I. (2018). Una perspectiva de la Calidad de Vida Laboral. Revista Iberoamericana de Ciencias, 5(6), 118-128.

La calidad de vida profesional (CVP) es el equilibrio de las demandas y recursos laborales/personales: es importante relacionar la CVP

Cabezas, C. (2000). La calidad de vida de los profesionales. Fmc, 7(Supl 7), 53-68.

Fernández Martínez O, García del Río García B, Hidalgo Cabrera C, López López C, Martín Tapia A, Moreno Suárez S. Percepción de la calidad de vida profesional de los médicos residentes de dos hospitales de distinto nivel asistencial. Medicina de Familia (And). 2007;2:11---8. 10.

Garrido S, García E, Viúdez I, López C, Más E, Ballarín M. Estudio de la calidad de vida profesional en trabajadores de atención primaria del Área 7 de la comunidad de Madrid. Rev Calid Asist. 2010;25:327---33

Minor limitation:
In line 67 References to burnout, key to quality of professional life.
83 it is suggested to add the direct contact and the high demands of patients in the care action (work stress and burnout).

Do not take for granted concepts or resources to be used. Detail more relevant issues such as Survey Monkey application, the reader does not need to know that it is a platform for online surveys. Possible limitations related to its use are described and taken into account (line 152).

Methodology.
Adequate quantitative methodology. Not only adequate, but each step is perfectly explained, allowing a replication of them.
The descriptive ones serve us to know the values more or less identified ()predominant) in the population under study or the correlations inform us of how the two variables (values and Quality of professional life) are related.

In a second step, we try to analyze whether there are significant differences in the various groups of respondents in terms of temporality/stability in employment, confirming the proposed working hypotheses.
The two questionnaires used have proven scientific validity and acceptance.
Adequate and well-explained procedure. The use of tables for the presentation of results is appreciated, well analyzed and well explained, making it easy to follow all the analyses carried out in a clear manner. Presentation of results Adequate presentation of the results; the tabular format, before so many different analyses, is ideal, making it easier to understand and follow the order and different results.

Suggestion in the methodological area: Continue the research line (further research) complementing it with qualitative methodologies (interviews and focus groups) more appropriate to measure the perceptions of the dimensions under study.
Findings consistent with other studies (nurses' value system and their QPL are influenced by their tenure).

Adequate distinction between discussion and conclusions; the discussion is correct but studies on values could be added (almost all satisfaction with quality of professional life, giving weight to the relationship between values and quality of professional life).

Suggest

Guerra-Martín, M. D. (2007). Priorización de valores enfermeros. Estudio de un distrito sanitario de Sevilla. Cultura de los cuidados, Año XI, n. 21 (1. semestre 2007); pp. 55-62.

Llorente-Alonso, M. A. R. T. A. FATIGA POR COMPASIÓN, VALORES Y EUTANASIA. SORIA, 7.

Writing
The writing is correct; clear, well-written and well-structured text. Good structure, good visualization of the information
In line 91 Specify that it is alluded to H1
Appropriate title and keywords.
Bibliographical references; it is suggested to complete the theoretical framework and some biographical suggestions that may or may not be used. In any case, it should be expanded in relation to the changes introduced by the authors.
Ethical guarantees in the execution of the article are adequately noted.

Author Response

Dear Reviewer 1. 

Thank you very much for all the contributions you have made. They certainly improve the quality of our manuscript. 
Thus, we have taken your suggestions into consideration. Most of the references you indicated have been included in order to improve the state of the art. A brief description of the Survey Monkey application has also been included.

You can review the changes made to the text (see in blue). 

In relation to the interest of study.
Novelty and opportunity.
Significant contributions of the research related to the attempt to determine the predominant values in health care and their potential change (specifically in the field of nursing) based on two major variables (1) employment status that speaks of stability or instability in employment and (2) seniority in the position. These variables are determinant and well related to the type of study; do the values change before specific situations linked to stability and seniority in the position?

Response: Thank you for your positive comments.

What relationship do they have with the quality of working life?

Response: Thank you for your meaningful observation. A text addressing this issue has been inserted at the beginning of the manuscript.

The research questions are logical and are well specified in objectives and working hypotheses.
The quantitative methodology used, although not new, is adequate.
The results of the study demonstrate the need for further research on the dimensions of values and quality of life as well as their impact on performance and quality of care in the field of health care.

State of the question

In the theoretical field, an important limitation is detected: it is necessary to explain what is understood by quality of professional life (explicit theoretical model and dimensions of analysis), as well as to better justify the relationship between quality of professional life and possession of values. It is advisable to better justify the relationship of the values assumed as proper to nursing with the proposed model of the universal values of Schwarz (they allude to culture, cognitive sphere of the person that impacts on his/her behavior).

Response: Thank you for your precise remarks. We have inserted a text at the beginning of the manuscript and also in the discussion that responds to your suggestion. Thank you very much for your references, we have used these manuscripts to reinforce this section of the article.

References that we can suggest

Wauters, M., Zamboni Berra, T., de Almeida Crispim, J., Arcêncio, R. A., & Cartagena-Ramos, D. (2023). Calidad de vida del personal de salud durante la pandemia de COVID-19: revisión exploratoria. Revista Panamericana de Salud Pública, 46, e30.

Vizcaíno, Y. Y. M., Alonso, M. D. C. V., & Vizcaíno, Y. M. (2020). Factores involucrados en la calidad de vida laboral para el ejercicio de la enfermería. Revista Cubana de Medicina Militar, 49(2), 364-374.

Castro, P., Cruz, E., Hernández, J., Vargas, R., Luis, K., Gatica, L., & Tepal, I. (2018). Una perspectiva de la Calidad de Vida Laboral. Revista Iberoamericana de Ciencias, 5(6), 118-128.

La calidad de vida profesional (CVP) es el equilibrio de las demandas y recursos laborales/personales: es importante relacionar la CVP

Cabezas, C. (2000). La calidad de vida de los profesionales. Fmc, 7(Supl 7), 53-68.

Fernández Martínez O, García del Río García B, Hidalgo Cabrera C, López López C, Martín Tapia A, Moreno Suárez S. Percepción de la calidad de vida profesional de los médicos residentes de dos hospitales de distinto nivel asistencial. Medicina de Familia (And). 2007;2:11---8. 10.

Garrido S, García E, Viúdez I, López C, Más E, Ballarín M. Estudio de la calidad de vida profesional en trabajadores de atención primaria del Área 7 de la comunidad de Madrid. Rev Calid Asist. 2010;25:327---33

Minor limitation:
In line 67 References to burnout, key to quality of professional life.

Response: Done. Thank you for your suggestions.

83 it is suggested to add the direct contact and the high demands of patients in the care action (work stress and burnout).

Response: Done. Thank you for your suggestions.

Do not take for granted concepts or resources to be used. Detail more relevant issues such as Survey Monkey application, the reader does not need to know that it is a platform for online surveys. Possible limitations related to its use are described and taken into account (line 152).

Response: Done. The text has been amended and it has been explained that it is an online platform for surveys.

Methodology.
Adequate quantitative methodology. Not only adequate, but each step is perfectly explained, allowing a replication of them.
The descriptive ones serve us to know the values more or less identified ()predominant) in the population under study or the correlations inform us of how the two variables (values and Quality of professional life) are related.

Response: Thank you for your positive comments.

In a second step, we try to analyze whether there are significant differences in the various groups of respondents in terms of temporality/stability in employment, confirming the proposed working hypotheses.
The two questionnaires used have proven scientific validity and acceptance.
Adequate and well-explained procedure. The use of tables for the presentation of results is appreciated, well analyzed and well explained, making it easy to follow all the analyses carried out in a clear manner. Presentation of results Adequate presentation of the results; the tabular format, before so many different analyses, is ideal, making it easier to understand and follow the order and different results.

Suggestion in the methodological area: Continue the research line (further research) complementing it with qualitative methodologies (interviews and focus groups) more appropriate to measure the perceptions of the dimensions under study.
Findings consistent with other studies (nurses' value system and their QPL are influenced by their tenure).

Response: Thank you very much for your very helpful comments and suggestions.

Adequate distinction between discussion and conclusions; the discussion is correct but studies on values could be added (almost all satisfaction with quality of professional life, giving weight to the relationship between values and quality of professional life).

Response: Thank you very much for your comments. New contributions have been added to the discussion.

Suggest

Guerra-Martín, M. D. (2007). Priorización de valores enfermeros. Estudio de un distrito sanitario de Sevilla. Cultura de los cuidados, Año XI, n. 21 (1. semestre 2007); pp. 55-62.

Llorente-Alonso, M. A. R. T. A. FATIGA POR COMPASIÓN, VALORES Y EUTANASIA. SORIA, 7.

Writing
The writing is correct; clear, well-written and well-structured text. Good structure, good visualization of the information
In line 91 Specify that it is alluded to H1
Appropriate title and keywords.
Bibliographical references; it is suggested to complete the theoretical framework and some biographical suggestions that may or may not be used. In any case, it should be expanded in relation to the changes introduced by the authors.
Ethical guarantees in the execution of the article are adequately noted.

Response: Thank you very much for your very helpful comments and suggestions.

King regards

Reviewer 2 Report

First sentence line 31. It would be much less confusing to says 'Quality of Professional Life'

lines 93-97. I think these hypotheses could be made clearer, because you definitely expect the relationship to be in a particular direction and QPL to be a dependent variable in H2 and H3 (this directionality becomes clear in your Discussion section).  For example,

H2. The terms of employment of nursing staff influence their QPL and their axiological profile.

H3. Tenure of service among nursing staff influences their QPL.

Viewed in this way, it seems that there may need to be a fourth hypothesis relating tenure of service and social values.

Line 178 - axiological, not axiomatic

Line 208 Under table 3, you give *p < .01. **p < .05. These are the wrong way round. The same is true for line 217

Author Response

Dear Reviewer 2.

Thank you very much for all the contributions you have made. They certainly improve the quality of our manuscript.
Thus, we have taken your suggestions into consideration.

The wording of the hypotheses establishes a relationship of occurrence between the factors measuring quality of professional life, values and working conditions. However, we do not establish a causal relationship between these variables, nor do we claim that the results in one are a result of the others.
The relationship between length of service and values is a very interesting line of research that raises the need to analyse whether a longer time in nursing is related to a greater identification with the prosocial values of nursing. However, this is not a hypothesis that can be put forward in the present research as there is no data available for its analysis.

You can review the changes introduced in the text (see in blue).

First sentence line 31. It would be much less confusing to says 'Quality of Professional Life'

Response: Thank you very much for your input.

lines 93-97. I think these hypotheses could be made clearer, because you definitely expect the relationship to be in a particular direction and QPL to be a dependent variable in H2 and H3 (this directionality becomes clear in your Discussion section).  For example,

H2. The terms of employment of nursing staff influence their QPL and their axiological profile.

H3. Tenure of service among nursing staff influences their QPL.

Response: Thank you very much for your suggestion. The wording of the hypotheses establishes a relationship of occurrence between the factors measuring quality of professional life, values and working conditions. However, we do not establish a causal relationship between these variables, nor do we claim that the results in one are a result of the others.
The relationship between length of service and values is a very interesting line of research that raises the need to analyse whether a longer time in nursing is related to a greater identification with the prosocial values of nursing. However, this is not a hypothesis that can be put forward in the present research as there is no data available for its analysis. 

Viewed in this way, it seems that there may need to be a fourth hypothesis relating tenure of service and social values.

Line 178 - axiological, not axiomatic

Response: Done. Thank you for your precise remarks.

Line 208 Under table 3, you give *p < .01. **p < .05. These are the wrong way round. The same is true for line 217

Response: Done. Thank you for your precise remarks.

King regards
